# Variational autoencoder-based chemical latent space for large molecular structures with 3D complexity

Toshiki Ochiai[1], Tensei Inukai[1], Manato Akiyama[1], Kairi Furui [2], Masahito Ohue [2], Nobuaki Matsumori [3], Shinsuke Inuki[4], Motonari Uesugi [5], Toshiaki Sunazuka[6], Kazuya Kikuchi[7,8], Hideaki Kakeya [4] & Yasubumi Sakakibara [1,9✉]

The structural diversity of chemical libraries, which are systematic collections of compounds that have potential to bind to biomolecules, can be represented by chemical latent space. A chemical latent space is a projection of a compound structure into a mathematical space based on several molecular features, and it can express structural diversity within a compound library in order to explore a broader chemical space and generate novel compound structures for drug candidates. In this study, we developed a deep-learning method, called NP-VAE (Natural Product-oriented Variational Autoencoder), based on variational autoencoder for managing hard-to-analyze datasets from DrugBank and large molecular structures such as natural compounds with chirality, an essential factor in the 3D complexity of compounds. NP-VAE was successful in constructing the chemical latent space from large-sized compounds that were unable to be handled in existing methods, achieving higher reconstruction accuracy, and demonstrating stable performance as a generative model across various indices. Furthermore, by exploring the acquired latent space, we succeeded in comprehensively analyzing a compound library containing natural compounds and generating novel compound structures with optimized functions.

[1] Department of Biosciences and Informatics, Keio University, Yokohama, Kanagawa 223-8522, Japan. [2] Department of Computer Science, School of Computing, Tokyo Institute of Technology, Yokohama, Kanagawa 226-8501, Japan. [3] Department of Chemistry, Graduate School of Science, Kyushu University, Fukuoka, Fukuoka 819-0395, Japan. [4] Division of Medicinal Frontier Sciences, Graduate School of Pharmaceutical Sciences, Kyoto University, Kyoto, Kyoto 606-8501, Japan. [5] Institute for Chemical Research and WPI-iCeMS, Kyoto University, Uji, Kyoto 611-0011, Japan. [6] Omura Satoshi Memorial Institute and Graduate School of Infection Control Sciences, Kitasato University, Minato-ku, Tokyo 108-8641, Japan. [7] Department of Applied Chemistry, Graduate School of Engineering, Osaka University, Suita, Osaka 565-0871, Japan. [8] Immunology Frontier Research Centre, Osaka University, Suita, Osaka 565-0871, Japan. [9] Department of Data Science, Kitasato University School of Frontier Engineering, Sagamihara, Kanagawa 252-0373, Japan. ✉email: yasu@bio.keio.ac.jp

It is estimated that there are approximately $10^{60}$ variations in all possible compound structures, even when limited to small molecules with a molecular weight of 500 or less[1]. Structural diversity within compound libraries is crucial for discovering new pharmaceutical compounds, necessitating coverage of as many candidates as possible from this vast pool. The structural diversity of compounds in a compound library can be represented by a chemical latent space, which projects compound structures onto a mathematical space based on various molecular features, such as fingerprints. Building a chemical latent space that potentially contains numerous unknown compound structures beyond existing libraries is an important research topic in cheminformatics. On the other hand, many of the natural compounds produced by living organisms have complex and unique structures compared to conventional drugs, and they exhibit high biological activity[2]. By applying state-of-the-art generative artificial intelligence techniques to the heterogeneous data from both the approved drug database and natural product compound libraries, it becomes possible to virtually generate and design novel compound structures that combine the characteristics of both types of data. This approach has become a global trend in cutting-edge drug discovery, known as DMTA (design-make-test-analyze)[3].

Various deep-learning models have been developed with the aim of constructing chemical latent spaces and computationally generating new compound structures. Variational autoencoder (VAE)[4] is one of the representative deep learning methods for constructing chemical latent spaces. VAE consists of two components: an encoder, which transforms input values into low-dimensional variables called latent variables, and a decoder, which transforms latent variables into corresponding output values. By exploring the chemical latent space, unknown compound structures not present in the training data could be generated. VAEs that handle compound structures as inputs and outputs are broadly classified into two types: SMILES-based and graph-based. Chemical VAE (CVAE)[5], the earliest model of SMILES-based methods, takes SMILES strings[6], which represent compound structures as strings, as input and constructs a latent space by projecting a compound library into a low-dimensional space. CVAE was a pioneering study that applied VAE for constructing a latent space of chemical compounds. However, CVAE, which outputs SMILES strings symbol by symbol, faces the issue that most of the output does not conform to chemical rules and fails to form a valid compound structure. Therefore, CVAE added an ad-hoc one-step process to validate the chemical structures output from the decoder and discarded the invalid ones. Grammar VAE (GVAE)[7] and Syntax-Directed VAE (SD-VAE)[8] were developed as models capable of generating more valid compound structures by focusing on the grammatical structure of SMILES strings. Recently, studies[9–11] applying the SMILES representation have again become active as chemical language model (CLM). These state-of-the-art methods adopt recurrent neural networks (RNNs) such as LSTM[12] to learn the SMILES grammar using a large amount of pretraining data and employ transfer learning to the compound structures of interest. Nevertheless, the SMILES representation-based model suffers from generating invalid SMILES strings, and hence requires the filtering out of these invalid outputs[13]. In addition, one crucial difference compared with the VAE approach is that these RNN-based models are merely generative and do not explicitly construct a vector space (latent space) and its latent variables that are projected from and can be reverse-mapped to the actual compounds, which means they do not provide the capability to explore the latent space of structurally diverse compounds in the library.

Graph-based models include Constrained Graph VAE (CG-VAE)[14] and Junction Tree VAE (JT-VAE)[15]. These models represent compound structures as graph structures defined by adjacency relationships between atoms, enabling them to generate completely valid outputs. Specifically, JT-VAE achieved high reconstruction accuracy by treating molecular graphs not only as graph structures but also as tree structures. However, these models were all designed for small molecules and could not handle large compound structures due to their high spatial order. As a result, Hierarchical VAE (HierVAE)[16] was developed to apply VAE to larger compound structures. By handling compound structures in relatively large fragment units, HierVAE demonstrated high accuracy for datasets composed of large compounds with repeating structures. Nevertheless, challenges remain, such as the inability to consider stereochemistry and the difficulty in applying the model to massive and complex compound structures with diverse internal structures like natural compounds. It is worth noting that natural compounds are quite different from massive compounds with uniform internal structures, like polymers.

Another state-of-the-art generative model is the flow-based model[17], which uses deep learning to project and generate compound structures. Flow-based models map to a latent space that guarantees inverse transformation by repeatedly applying invertible functions to input data. In other words, the reconstruction accuracy of flow-based models is always guaranteed to be 100%, regardless of the degree of learning. The graph-based flow model, MoFlow[18], represents compound structures as two types of bit matrices and can acquire completely invertible latent variables by applying a normalizing flow to each. However, this does not necessarily indicate the accuracy or continuity of the resulting latent space, unlike the case with VAEs. Furthermore, Flow-based models have a problem that the latent space becomes high-dimensional in order to guarantee reversibility. Since a latent space with the same dimension as the input dimension is constructed, the space exploration is highly harder than with VAEs, which project onto a lower-dimensional space. Moreover, flow-based learning can be quite unstable, making it prone to gradient explosion when the input dimension increases. Therefore, when using flow-based models for compound data, it is limited to compounds with a small size of input compounds.

As discussed above, deep learning models developed thus far have struggled to effectively handle large, complex and heterogeneous compound structures, such as natural compounds. Natural compounds produced by organisms such as microbes and plants often possess novel structures, and due to their characteristics of being produced during biological processes, many of them exhibit high biological activity[2,19]. Indeed, natural products have been widely used as drugs, such as antibiotics like Penicillin and Streptomycin, and anticancer agents like Bryostatin and Epothilone[20]. Therefore, constructing a chemical latent space from a compound library that includes natural compounds plays a crucial role in drug discovery.

In this study, we developed a VAE-based deep learning method, called Natural-Product Compound Variational Autoencoder (NP-VAE), to handle compounds with complex molecular structures like natural compounds and acquire a chemical latent space that projects large molecular structures. NP-VAE is a graph-based VAE that combines an algorithm for effectively decomposing compound structures into fragment units and converting them into tree structures, along with the Extended Connectivity Fingerprints (ECFP)[21], and the Tree-LSTM[22], a type of recurrent neural network. NP-VAE, which has 12 million parameters, was successfully developed through significant improvements to the algorithms of JT-VAE[15] and HierVAE[16].

The first objective of this study is to obtain a highly interpretable chemical latent space that includes middle/large molecular structures like natural compounds using NP-VAE. We construct a latent space

**Table 1 Performance comparison of generalization ability between NP-VAE and existing methods (baseline models).**

|         | 2D Reconstruction accuracy (test) | Validity |
|---------|-----------------------------------|----------|
| NP-VAE  | 0.813 | 1.000 |
| HierVAE | 0.799 | 1.000 |
| JT-VAE  | 0.585 | 1.000 |
| CG-VAE  | 0.424 | 1.000 |
| CVAE    | 0.215 | 0.931 |

The reconstruction accuracy and validity for test compounds in the evaluation dataset were compared.
The reconstruction accuracy and validity of existing methods were taken from values reported in the previous study[16].

**Table 2 Comparison of the maximum number of atoms and molecular weight of the compounds included in the dataset.**

|                           | Maximum number of atoms | Maximum molecular weight |
|---------------------------|-------------------------|--------------------------|
| Drug-and-natural-product dataset | 551 | 8272 |
| Restricted dataset        | 100 | 1626 |
| DrugBank                  | 551 | 8272 |
| Project dataset           | 457 | 6574 |
| ZINC                      | 38  | 500  |

In the maximum number of atoms, only non-hydrogen atoms were counted. Compared to the ZINC dataset used in previous studies, the drug-and-natural-product dataset (combined drug and natural product dataset of DrugBank and the project dataset) includes compounds with more than 13 times larger molecular weights.

that includes hard-to-analyze DrugBank[23] and a large collection of natural compounds, which previous studies could not handle, and effectively perform statistical and comprehensive functional analysis of compound libraries. Furthermore, NP-VAE is developed to deal with chirality, which is an essential factor in the 3D structure of compounds. The second objective is to generate novel compound structures optimized for the target function (property) by exploring the acquired latent space. NP-VAE has a mechanism to train the chemical latent space incorporating functional information along with structural information. The obtained chemical latent space enables the design of optimized compound structures as molecular-targeted drugs by generating new compounds from the surrounding sub-space of an existing pharmaceutical drug, such as anticancer drugs. Furthermore, by combining NP-VAE to generate novel compound structures with docking analysis, we demonstrate the usefulness of this method as an *in-silico* drug discovery tool.

## Results and discussions

**Performance evaluation of NP-VAE as reconstruction and generative model.** First, we evaluated the reconstruction accuracy of the proposed NP-VAE for test compounds that were not included in the training compound data, which is referred to as the generalization ability. Evaluating the generalization ability is crucial because it allows us to verify how accurately the constructed chemical latent space has been interpolated. Following the study HierVAE[16] that conducted an evaluation of the generalization ability, we used St. John et al.'s dataset[24] (hereinafter referred to as the evaluation dataset). This dataset was divided into 76,000 training compounds, 5000 validation compounds, and 5000 test compounds, exactly same as the previous study[16]. After training on the training compounds, the reconstruction accuracy and validity for test compounds were calculated. Reconstruction accuracy was determined using the Monte Carlo method for 5000 test compounds. In other words, for each test compound, 10 encodings were performed, and 10 decodings were conducted for each encoding, resulting in 100 output compounds. The proportion of compound structures that matched between the input to the encoder and the output from the decoder was calculated. To determine validity, 1000 latent vectors were sampled from the prior distribution $N(0, I)$, and after decoding each of them 100 times, the proportion of chemically valid output compounds was examined using RDKit[25]. Four state-of-the-art compound VAEs, namely CVAE[5], CG-VAE[14], JT-VAE[15], and HierVAE[16], were compared as baseline models. As shown in Table 1, NP-VAE demonstrated higher reconstruction accuracy for the test compounds compared to the previous models. Additionally, since NP-VAE generates compounds in substructure units (fragments) rather than single-atom units, the generation success rate is always 100%. These

results indicate that NP-VAE is a high-performance generative model, suggesting that the chemical latent space constructed by NP-VAE contains sufficient information to accurately estimate unknown compounds from known compounds.

Next, we compared the performance of NP-VAE as generative model with the state-of-the-art generative models, the Flow-based model MoFlow[18], and the SMILES-based method employing CharRNN (character-level RNN)[26], which is also referred to as SM-RNN and demonstrated high performance in the study[10], and the VAE model HierVAE.

Since our primary motivation is to develop a VAE model capable of handling large and complex molecules for the construction of chemical latent space, we prepared a compound library consisting of approximately 30,000 compounds. This library combines around 10,000 compounds from DrugBank, a public database containing numerous approved compounds, and approximately 20,000 compounds from the project dataset (hereinafter referred to as the drug-and-natural-product dataset). The project dataset is an original compound library collected from various laboratories through the Ministry of Education, Culture, Sports, Science, and Technology-designated project, "Frontier Research on Chemical Communications"[27], in which this research participated. The project dataset mainly includes natural compounds, and compared to compounds in the frequently used ZINC database[28], it contains a number of complex and large molecules (see Supplementary Fig. S1 for illustration). However, the state-of-the-art VAE models, JT-VAE[15], and HierVAE[16], and the flow-based model MoFlow[18] were unable to handle data for compounds of this larger size, so we had to prepare a restricted dataset where all existing methods can be executed. This restricted dataset (hereinafter referred to as the restricted dataset) was constructed by first reducing the drug-and-natural-product dataset to compounds with fewer than 100 non-hydrogen atoms and further removing some compounds that caused errors with HierVAE. Therefore, initially, we compared the difference in maximum compound size between the drug-and-natural-product dataset and the restricted dataset. The Table 2 shows the comparison of the maximum number of atoms and molecular weight of the compounds included in the drug-and-natural-product dataset, the restricted dataset, and three other databases. Note that regarding JTVAE, it took an impractical amount of computation time and did not complete the calculations even with the restricted dataset; therefore, it was excluded from all subsequent experiments.

Next, we performed the process of randomly sampling 5000 latent vectors from the prior Gaussian distribution $N(0, I)$ five times for each model. Then, we calculated and compared the following variety of metrics proposed in benchmarking studies such as MOSES[29] and GuacaMol[30].

**Table 3 Comparison of NP-VAE, HierVAE, MoFlow and SM-RNN as generative models: For the generated compounds, 2D and 3D reconstruction accuracy, uniqueness, novelty, logP, SAscore, QED, Filters, SNN, molecular weight, NP-likeness, and the distance between compound property distributions, Frag, Scaf, IntDiv, and Phys div were calculated.**

|  | NP-VAE | HierVAE | MoFlow | SM-RNN |
|---|---|---|---|---|
| 2D Reconstruction accuracy | 0.871 | 0.438 | **1.000** | N/A |
| 3D Reconstruction accuracy | **0.853** | N/A | N/A | N/A |
| Uniqueness | 0.981 ± 0.003 | **0.987** ± 0.002 | 0.970 ± 0.002 | 0.939 ± 0.002 |
| Novelty | 0.983 ± 0.002 | 0.991 ± 0.001 | **0.998** ± 0.000 | 0.223 ± 0.010 |
| logP | 2.255 ± 1.795 | 2.445 ± 1.815 | 1.417 ± 1.433 | **2.654** ± 2.193 |
| QED | **0.670** ± 0.025 | 0.619 ± 0.038 | 0.314 ± 0.017 | 0.566 ± 0.050 |
| SAscore | **2.378** ± 0.551 | 3.021 ± 0.797 | 4.370 ± 1.391 | 3.052 ± 1.069 |
| Filters | **0.824** ± 0.002 | 0.768 ± 0.008 | 0.263 ± 0.003 | 0.777 ± 0.005 |
| SNN | 0.484 ± 0.003 | 0.410 ± 0.003 | 0.304 ± 0.003 | **0.920** ± 0.004 |
| MolWt | 255.98 ± 85.64 | 327.41 ± 116.64 | 196.83 ± 79.42 | 375.24 ± 148.10 |
| NP-likeness | -0.758 ± 0.832 | -0.672 ± 0.896 | **0.887** ± 0.656 | -0.667 ± 1.186 |
| Frag | 0.951 ± 0.001 | 0.987 ± 0.001 | 0.410 ± 0.008 | **0.999** ± 0.000 |
| Scaf | 0.444 ± 0.015 | 0.307 ± 0.015 | 0.000 ± 0.000 | **0.810** ± 0.013 |
| IntDiv | 0.877 ± 0.001 | 0.881 ± 0.001 | **0.903** ± 0.001 | 0.887 ± 0.000 |
| Phys div | 0.676 ± 0.007 | 0.843 ± 0.007 | 0.357 ± 0.003 | **0.955** ± 0.001 |

The highest value for each accuracy indices is shown in bold.

-Uniqueness: The proportion of unique molecules among the generated valid molecules. This serves as an indicator of the uniqueness of the generated compound structures, and the value will be low if the model has collapsed and generated only a few typical molecules.

- Novelty: The proportion of generated molecules that do not exist in the training set. A low value may indicate overfitting.

- LogP: Represents the lipophilicity of a molecule. A moderate lipophilicity is required for pharmaceutical compounds.

- QED: An indicator representing the drug-likeness of a molecule. Since it is calculated based on existing oral drugs, it can be considered an indicator of oral drug-likeness. It is expressed as a value between 0 and 1, with values closer to 1 indicating structures that are more like oral pharmaceuticals[31].

- SAscore: A score representing the difficulty of synthesis based on molecular structure. It is expressed as a value between 1 and 10, with values closer to 10 indicating higher synthesis difficulty[32].

- Filters: Represent the proportion of generated molecules that passed through a filter, which eliminates undesired structures used during the construction of the MOSES dataset[29]. A lower value indicates that there are more molecules with abnormal structures.

- SNN (similarity to a nearest neighbor): The average Tanimoto coefficient between each generated molecule and the most similar molecule within the training data. This value decreases as the generated molecules deviate further from the distribution of the training data.

- MolWt: Molecular weight.

- NP-likeness: A measure of naturalness. NP-likeness score is a measure designed to estimate how closely a given molecule resembles known natural products[33].

- Frag: Comparison of the distribution of BRICS fragmentations between generated molecules and the training data. The value increases when both sets contain molecules with similar fragments.

- Scaf: Comparison of the distribution of primary structural elements within molecules, referred to as scaffolds. Frag and Scaf are both metrics used to measure the similarity between generated molecules and the training data at the level of substructure units.

- IntDiv: This is an evaluation of the structural diversity within the set of generated molecules. It is calculated by computing the Tanimoto coefficient between all pairs of generated molecules and taking the average.

- Phy div: KL divergence between generated molecules and the training data in terms of physicochemical properties and is calculated from physicochemical descriptors such as molecular weight, the number of aromatic rings, and the count of rotatable bonds[30]. A higher value indicates better performance.

The results are shown in Table 3. First, the reconstruction accuracy for training compounds was compared among three models: NP-VAE, HierVAE, and MoFlow, which constitute the latent space using the encoder and decoder. This reconstruction accuracy indicates how accurately the compound library can be projected without information loss. Regarding the 2D reconstruction accuracy for the planar structure of compounds, NP-VAE significantly outperformed HierVAE. On the other hand, MoFlow achieved a 100% 2D reconstruction accuracy. From the mechanism of flow-based models, the reconstruction accuracy of flow-based models is always guaranteed to be 100%, regardless of the degree of learning. However, this does not necessarily indicate the accuracy or continuity of the resulting latent space, unlike the case with VAEs. Flow-based models have a problem that the latent space becomes high-dimensional in order to guarantee reversibility. Since a latent space with the same dimension as the input dimension is constructed, the space exploration is highly harder than with VAEs, which project onto a lower-dimensional space. Moreover, flow-based learning can be quite unstable, making it prone to gradient explosion when the input dimension increases. The 3D reconstruction accuracy, considering not only the planar structure but also the stereochemistry (chirality), of NP-VAE exceeded 85%, while the other two models could not handle the 3D structures, and hence, their accuracy was not available. This result demonstrates that NP-VAE has high performance as a feature extractor.

Second, NP-VAE demonstrated stable performance as a generative model across almost all indices. In terms of uniqueness, novelty and logP, NP-VAE showed comparable performance to the top-performing models. Due to the large variance in logP, the difference in logP scores among models is not statistically significant. NP-VAE exhibited the highest QED, SA score, and Filters score, which represents the drug-likeness of a molecule, the difficulty of its synthesis, and the proportion that passed through a filter to eliminate undesired structures,

respectively. On the other hand, SM-RNN exhibited a difference in performance values of nearly two times for SNN, Scaf and Phys div metrics; however, these results are not informative. The compound structures generated by SM-RNN had the Novelty of only 22%, which implies that 70% of the structures were identical to the training data. It is obvious that these three metrics plus Frag score would improve if the model outputs compound structures identical to the training data. In other words, SM-RNN is simply memorizing the training data, and its usefulness as a generative model for generating new structures is limited. Regarding molecular weight, we included a plot illustrating the size distributions of the molecules generated by all models in Supplementary Fig. S2. Each distribution is highly divergent, indicating the generation of diverse molecular weights. Regarding NP-likeness, MoFlow showed the highest value, being the only one with a positive score, while other models take negative values. This is attributed to the fact that MoFlow generates extremely long straight-chain structures, which are considered to have abnormal structures, as indicated by the low Filter score. Moreover, since these structures have a Scaf value of 0, meaning they completely lack a scaffold, they are considered to be natural-product-like due to their unusual and rare structures compared to general compounds. Therefore, the NP-likeness calculation method is prone to assigning higher values in such cases. On the other hand, the NP-likeness score of the training data is −0.638, mainly due to the inclusion of numerous approved drug compounds from DrugBank. This value is close to the negative NP-likeness scores shown by the other three models. In conclusion, NP-VAE successfully generates compound structures with the highest scores in desired metrics for drug-likeness, such as QED, SAscore and Filters, while maintaining similarity to the training data at the level of fragments and scaffolds, as indicated by Frag and Scaf values, and achieving high novelty of generated structures.

**Construction of chemical latent space with natural compounds**. We constructed two chemical latent spaces using the entire set of the drug-and-natural-product dataset: one obtained by training with only the structural information of the compounds and the other obtained by training with both structural information and the NP-likeness score[33], which serves as a measure of naturalness, as functional information. More precisely, the NP-likeness score is incorporated into the learning process as functional information, which is implemented by a component of the loss function. NP-likeness score is a measure designed to estimate how closely a given molecule resembles known natural products. It is based on the distribution of fragments (substructures) in the molecule compared to a reference set of known natural products[33]. A higher NP-likeness score suggests that a molecule is more 'natural product-like,' meaning that its structure is more similar to those of known natural products.

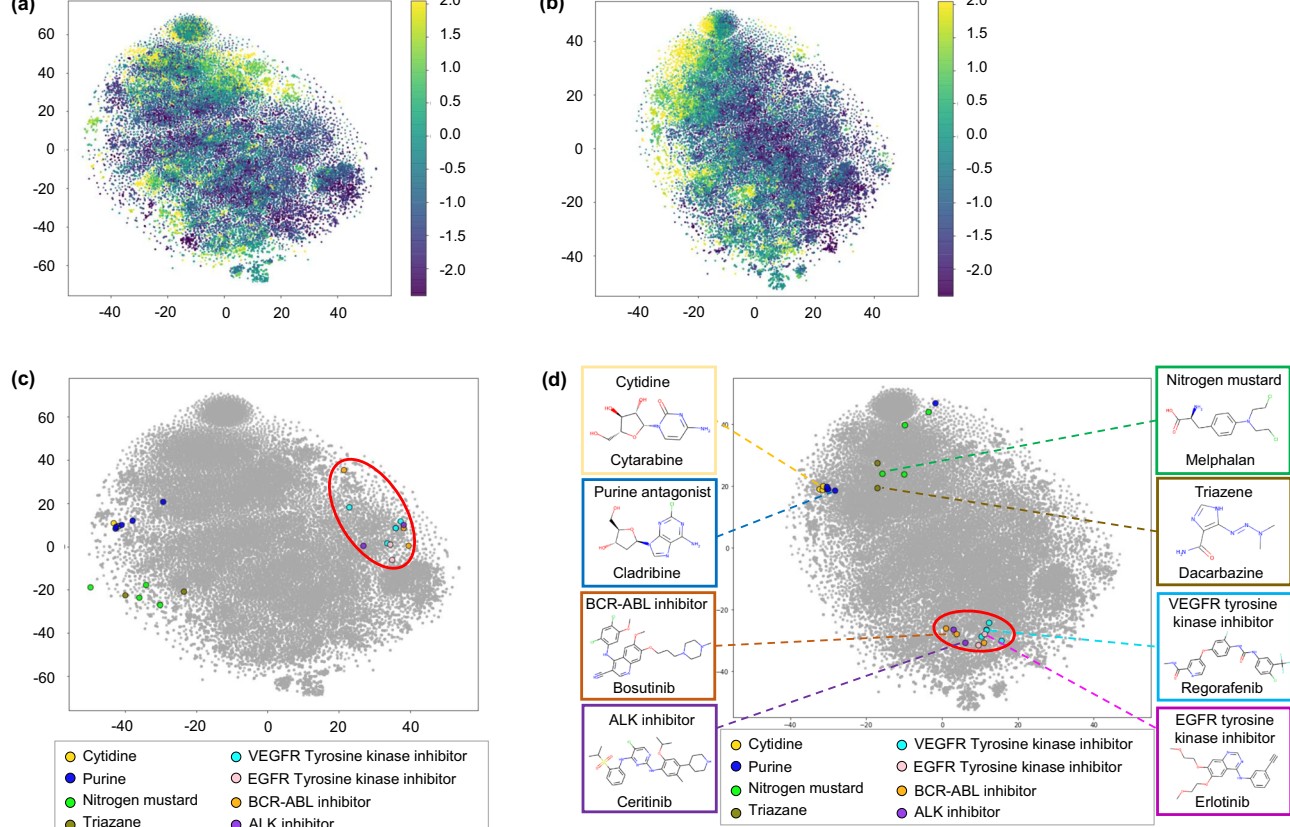

**Fig. 1 Visualization of chemical latent spaces using t-SNE plot. a, b** The higher the NP-likeness score, the more yellow it is, and the lower the score, the more purple it is. Compared to the chemical latent space (**a**) trained only on substructures, the chemical latent space (**b**) trained on both substructures and the NP-likeness score as functional information shows a more clustered distribution according to the NP-likeness score. Comparing the cases when plotting representative anticancer compounds in the chemical latent space trained only on substructures (**c**) and when plotting anticancer compounds in the chemical latent space trained on both substructures and NP-likeness scores (**d**), both show clustered distributions according to the anticancer drug classification, with chemotherapeutic drugs and molecular targeted drugs distributed separately. Focusing on the distribution of molecular targeted drugs (red frame), the distribution is more locally clustered when the NP-likeness score is included.

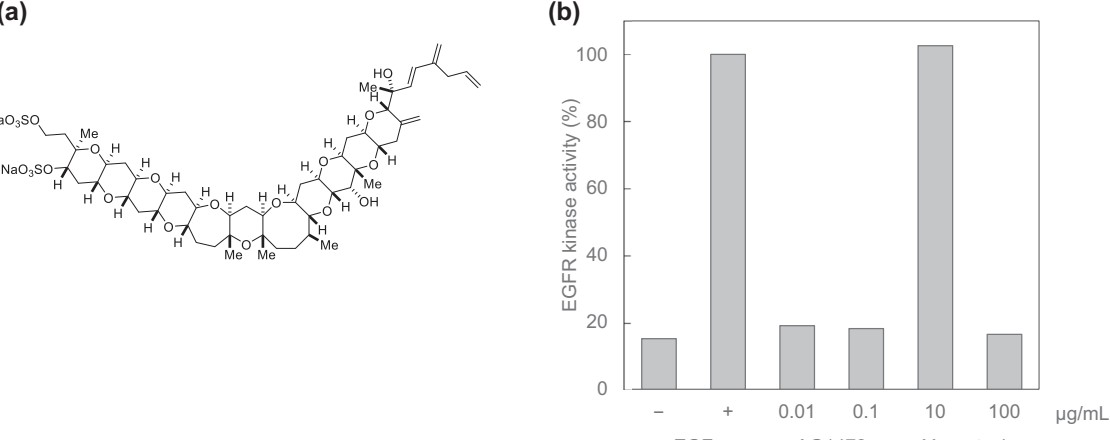

**Fig. 2 Yessotoxin and its EGFR inhibitory activity. a** Structure of Yessotoxin. Yessotoxin was first discovered in the 1980s from a scallop species called *Patinopecten yessoensis*[35] and since then, various derivatives have been found in crustaceans and algae[47]. **b** Inhibition of EGF-stimulated EGFR phosphorylation by Yessotoxin. EGFR tyrosine kinase activities are expressed as a percentage of the maximal phosphorylation induced by EGF. AG1478 is a selective EGFR inhibitor and was used as a positive control. When Yessotoxin was at 100 μg/ml, an inhibitory effect of over 80% was confirmed.

First, we visualized those two chemical latent spaces by reducing the dimension of latent variables to two dimension using t-SNE[34]. The results are shown in Fig. 1. In (a) and (b), compounds with higher NP-likeness scores are represented in yellow, while those with lower scores are depicted in purple. Comparing the latent space constructed using only the structural information of the compounds (Fig. 1a), the gradients of NP-likeness can be observed in the latent space constructed by incorporating NP-likeness scores as functional information (Fig. 1b). To quantitatively assess the gradients of NP-likeness in the chemical latent space, we calculated the correlation between the embedding distance and the difference of NP-likeness scores for randomly sampled pairs of points in the latent space. The results shown in Supplementary Fig. S3 indicates that the correlation (Pearson correlation coefficient $r = 0.19$) in the latent space incorporating NP-likeness scores is slightly higher than the correlation ($r = 0.14$) in the latent space constructed using only the structural information.

When plotting representative anticancer drug compounds on these chemical latent spaces, we observed more clustered distributions for each class of anticancer drugs in the space incorporating NP-likeness scores (Fig. 1d) compared to the space constructed with only structural information (Fig. 1c). In particular, many molecular-targeted drugs were found to be locally distributed. The reason for the localized distribution of existing molecular-targeted drugs in the space incorporating NP-likeness scores is considered to be due to the lower NP-likeness scores of molecular-targeted drugs compared to other pharmaceutical compounds (Supplementary Fig. S4). Thus, if a functional value for the drug of interest can be added during the training of NP-VAE as a functional indicator, a chemical latent space can be obtained where the desired pharmaceutical candidate compounds are locally distributed, and novel functionally optimized structures are easier to explore.

Second, by utilizing those advancement of the constructed chemical latent space, we found that the natural compound Yessotoxin (Fig. 2a)[35] included in the drug-and-natural-product dataset was located near existing molecular-targeted drugs. Based on this observation, we hypothesized that Yessotoxin isolated from a scallop species called *Patinopecten yessoensis* might have a function as a molecular-targeted drug. Upon conducting an assay, it was confirmed that Yessotoxin exhibited weak EGFR inhibitory activity (Fig. 2b). This suggests that exploring the chemical latent space constructed by

NP-VAE may also enable the discovery and annotation of unexpected compounds with similar pharmacological effects.

**Interpolation in chemical latent space**. For two existing drug compounds, we generated novel compound structures that exist between the two compounds by scanning the chemical latent space from one compound to the other. Let the latent variables of the starting and destination compounds be $z_s$ and $z_g$, respectively. When exploring $N$ equidistant points on the chemical latent space connecting the starting and destination points, the latent variable $z_i$ of the $i$-th generated compound was derived from the following equation.

$$z_i = z_s + \frac{z_g - z_s}{N} \cdot i \qquad (1)$$

By inputting this $z_i$ into the decoder, new compound structures can be generated. Figure 3 shows the novel compound structures obtained by exploring the space between two existing drugs, with the starting compound being a biomolecule, a Nicotinamide adenine dinucleotide derivative, and the destination compound being the molecular-targeted drug Sorafenib. As shown in Fig. 3, the similarity to the starting compound gradually decreased as one moves away from it and approaches the destination compound, while the similarity to the destination point compound gradually increased. Moreover, the NP-likeness score of the generated compound structures gradually decreased as the scanning progresses from a high to a low NP-likeness compound.

**Modification of compound structures by Bayesian optimization**. We used Bayesian optimization with TPE[36] to explore the chemical latent space and generate novel compound structures with optimized functional indicators. By setting an existing drug compound as the starting point and limiting the exploration range to the vicinity of the starting point in the chemical latent space, we generated novel compound structures with optimized functional indicators while maintaining structural similarity to the starting compound. The objective function to be maximized was set to the QED[31], an indicator of oral drug-likeness. The correlation coefficient between the NP-likeness score and QED in the drug-and-natural-product dataset is -0.31, indicating a negative correlation. Therefore, it is expected that there is an increasing gradient of QED in the space with an NP-likeness decreasing gradient, and the chemical latent space constructed in

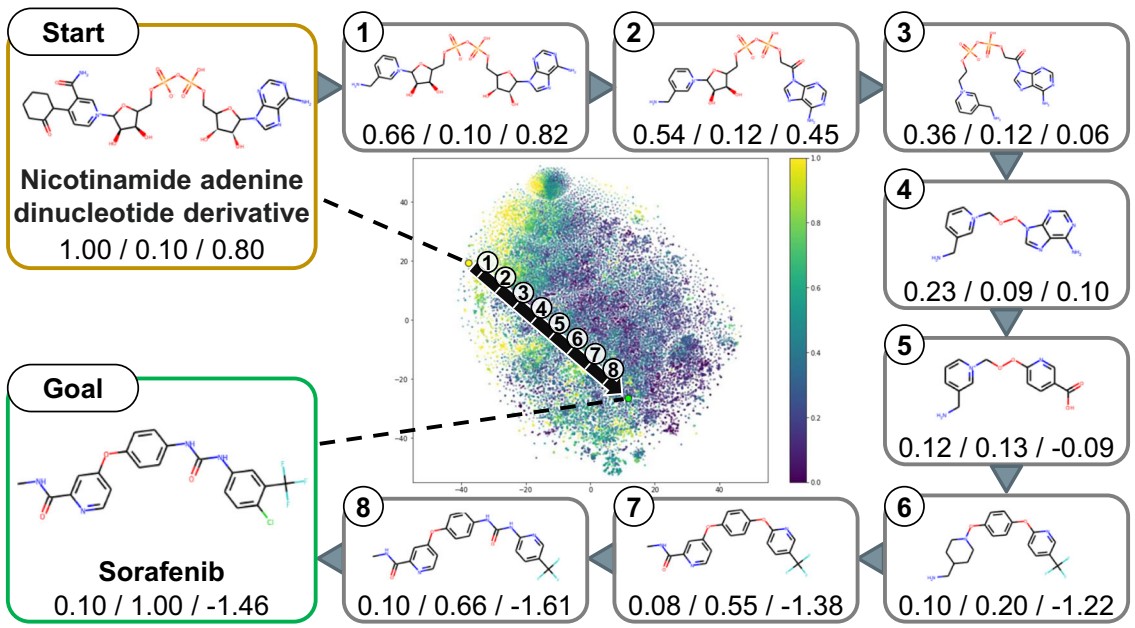

**Fig. 3 Generation of compound structures between two existing drugs through interpolation.** Interpolation of novel compound structures obtained by scanning the chemical latent space between two points, with the starting compound structure being a Nicotinamide adenine dinucleotide derivative from a biomolecule and the destination compound structure being Sorafenib, a molecular targeted drug. The three values below each compound structure represent, from left to right, the similarity to the starting compound, the similarity to the destination compound, and the NP-likeness score. As the compounds move closer to the destination point, the similarity to the starting compound gradually decreases, the similarity to the destination compound increases, and the NP-likeness score becomes lower.

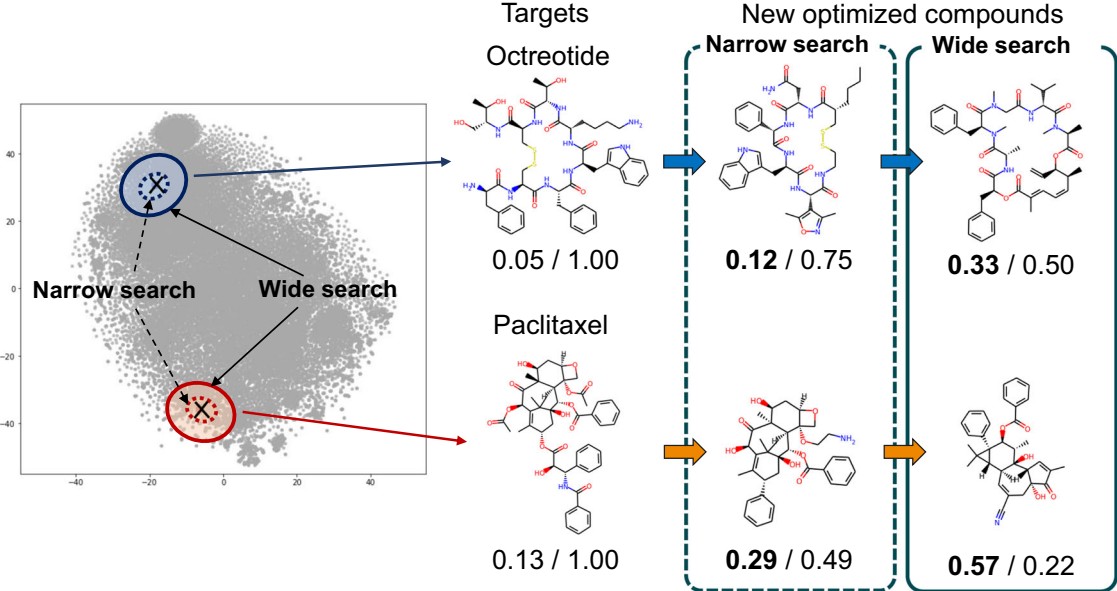

**Fig. 4 Generation of novel compound structures using Bayesian optimization.** The objective function to be maximized was set as the quantitative estimate of drug-likeness (QED), and novel compound structures with improved functional indices were explored using Bayesian optimization. The two values below each compound structure represent, from left to right, the QED score and the similarity to the starting compound. In this case, the search space was limited to the vicinity of the target compound, and optimization was performed in both narrow and wide search ranges, examining the effects on the resulting compound structures depending on the search space. When the search range was small, it was possible to obtain novel compound structures with improved QED while maintaining the characteristic structure of the target compound. When the search range was expanded, changes in the characteristic structure were observed, and novel compound structures with significantly improved QED could be obtained.

this study can also be used for exploring oral drug candidate compound structures.

Figure 4 shows the results of generating novel optimized compounds when the starting point for exploration is set to a peptide drug Octreotide and an anticancer drugs Paclitaxel.

When the exploration range is small, novel compound structures with optimized QED can be obtained while maintaining the characteristic structures of the target compounds. On the other hand, when the exploration range is expanded, although a bit large changes in the characteristic structures can be observed,

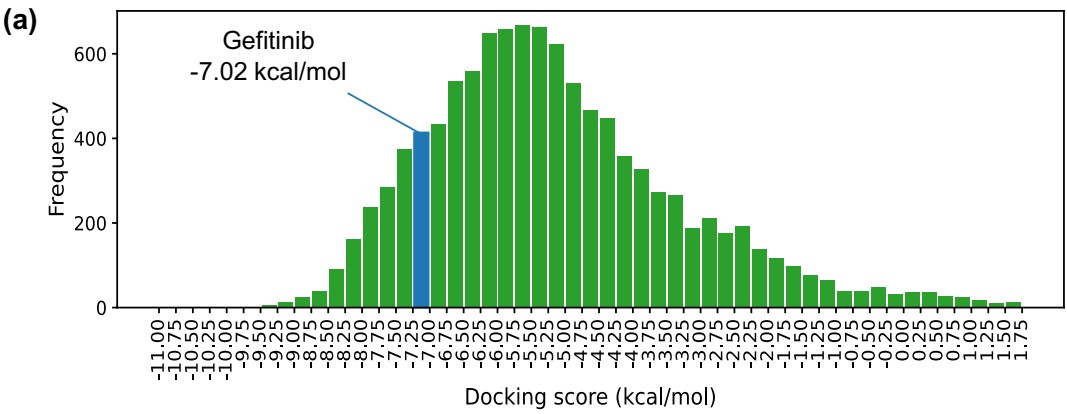

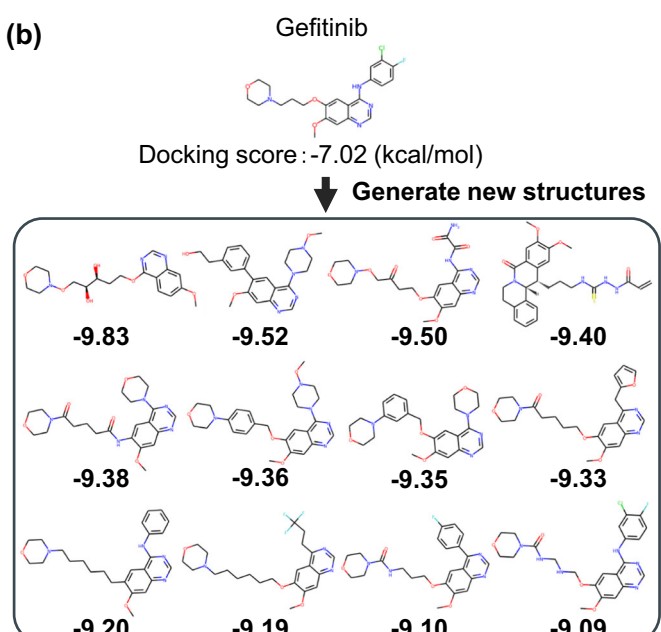

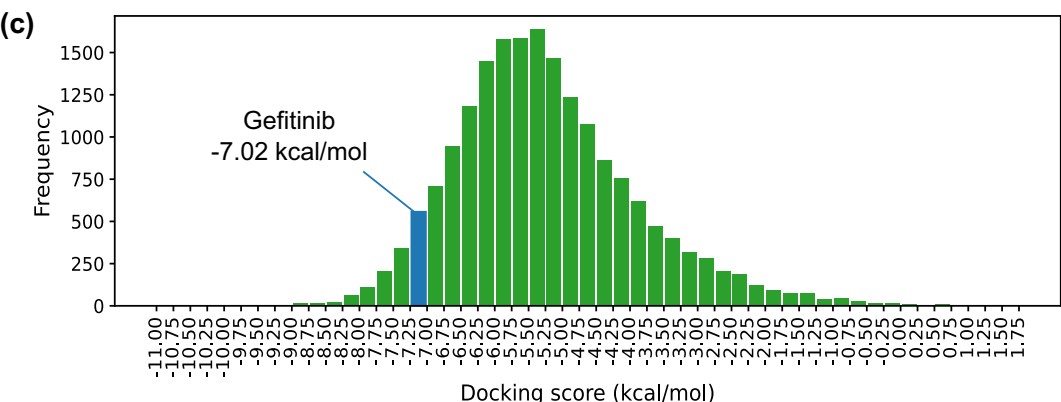

**Fig. 5 Generating novel compound structures from the vicinity of Gefitinib and calculating the docking scores with EGFR. a** Histogram with the number of generated compounds on the vertical axis and their docking scores on the horizontal axis. There were approximately 5700 novel compound structures with improved docking scores compared to Osimertinib, and about 1600 structures with improved scores compared to Gefitinib. **b** Novel generated compounds with top docking scores against EGFR. The numbers below the compound structures represent the docking scores. Among these, the majority of the structures contain a kinase-inhibiting quinazoline moiety, known to play a crucial role in EGFR interactions. In addition, it can be seen that the docking scores have been significantly improved due to the addition of other structural components. **c** Histogram of the docking scores for the virtual compounds generated by the machine-learning-based molecular generation tool, REINVENT (version 3.0)[42].

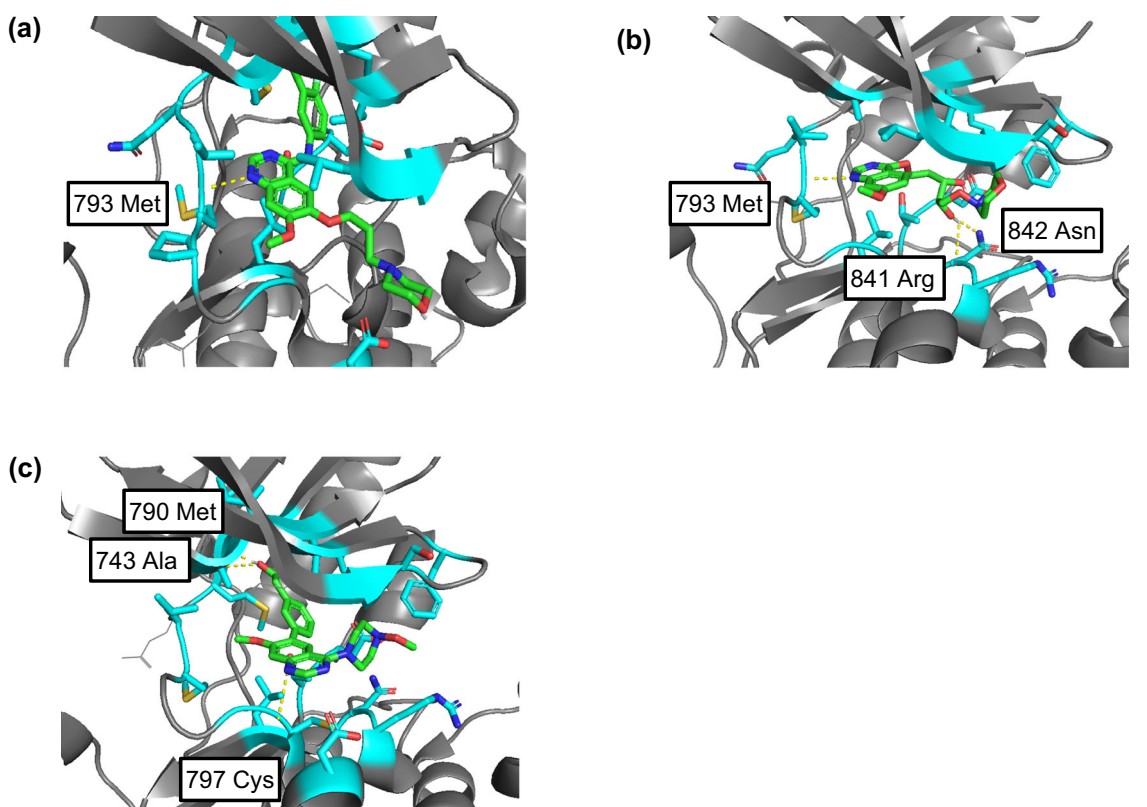

**Fig. 6 Docking poses between EGFR and Gefitinib, as well as EGFR and the novel generated compounds. a** Docking pose of interaction between EGFR and gefitinib, (**b**) Docking pose of interaction between EGFR and the novel compound with the highest docking score, and (**c**) docking pose of interaction between EGFR and the novel compound with the second-highest docking score. Carbon atoms within 4Å of the ligand compound in the EGFR structure are shown in light blue, and the parts where interactions were confirmed in the simulation results are indicated by yellow dashed lines. While Gefitinib is observed to interact with methionine at position 793, the ligand with the highest docking score was confirmed to interact with methionine at position 793, as well as arginine at position 841 and asparagine at position 842. Additionally, for the ligand with the second-highest docking score, interactions were observed with methionine at position 790, cysteine at position 797, and alanine at position 743.

**Table 4 Comparison of computational time between NPVAE and existing methods.**

|  | Computational time per epoch (sec) |
| --- | --- |
| NP-VAE | 1233.7 ± 3.84 |
| HierVAE | 1895.5 ± 51.73 |
| MoFlow | 305.1 ± 0.61 |
| SM-RNN | 48.8 ± 0.12 |

The computational time required for one epoch during training with the restricted dataset of drug-and-natural-product compounds is shown. The hardware specifications include Nvidia Tesla P100-SXM2, 16GB.

novel compound structures with significantly improved QED can be obtained. Natural product-derived drugs, such as Octreotide and Paclitaxel, are generally administered by injection. Therefore, improving the QED of such compounds is expected to enhance their properties as orally administered drugs, leading to increased convenience for patients. In addition, to quantitatively assess the effectiveness of Bayesian optimization, we repeated the optimization experiment for multiple points sampled from the latent space. When the exploration range was limited to compounds with a similarity of 0.6 or higher, the average improvement in the objective function QED was 0.046 with a standard deviation of 0.074. When the exploration range was expanded to compounds with a similarity of 0.2 or higher, the average improvement in the objective function QED significantly improved to 0.538 with a standard deviation of 0.022.

There are a few papers on functional optimization through sampling in the latent space. OptiMol proposed by Boitreaud et al.[37]. focused on an optimization strategy and targeted a specific aspect of drug discovery (binding affinities), whereas our NP-VAE model aims to deal with large, complex molecules with 3D structures and desired properties. The constrained Bayesian optimization method proposed by Griffiths et al.[38]. primarily used SMILES representations, which can lead to invalid outputs, making the handling of such outputs a significant focus of their work. In contrast, NP-VAE model is designed to effectively decompose the input compound structures into fragment units and convert them into tree structures to handle large and complex 3D molecular structures. Chembo proposed by Korovina et al.[39]. aimed to introduce a method for synthesizable recommendations beyond the SA score. The proposed Gaussian process-based approach has enough potential to be incorporated into functional optimization in NP-VAE.

**Generation of drug candidates from chemical latent space with docking analysis**. In the constructed chemical latent space, we generated approximately 10,000 novel compound structures from the vicinity of existing anticancer drug compounds. By performing docking analysis for the generated compound structures with the target proteins that interact with the original anticancer drug compounds, we searched for novel compound structures that are expected to have greater efficacy as molecular-targeted drugs. Schrödinger Glide[40] was used as the docking analysis software. When generating novel compound structures from the

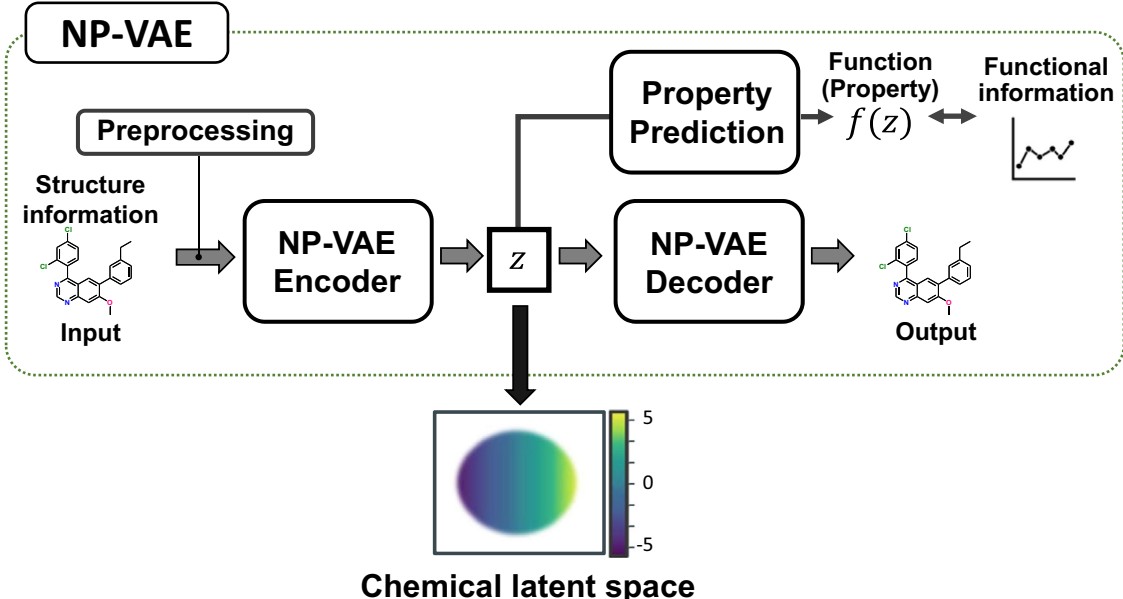

**Fig. 7 Overall structure of NP-VAE.** When the compound structure information is input to the Encoder, the latent variable $z$ is calculated based on the tree structure obtained by preprocessing. In the Decoder, the compound structure is calculated and output using a continuous algorithm with $z$ as the input. During training, a pathway is used in parallel to predict the functional indices of the compound with the latent variable as input. This allows for the construction of a chemical latent space that takes into account not only structural information but also functional information.

space surrounding molecular-targeted drugs such as Gefitinib and Osimertinib, we successfully discovered multiple compound structures with significantly better docking scores than the original compounds, as shown in Fig. 5a. There were about 5700 novel compound structures with improved docking scores compared to Osimertinib and about 1600 structures with improved scores compared to Gefitinib. Figure 5b shows the top-ranking compound structures in terms of docking scores. Many of these structures share the pyrimidine moiety, which is known to play an important role in the interaction with EGFR[41], with Gefitinib. However, the docking scores are greatly improved by the addition of other structures. These results suggest that, in the chemical latent space constructed by NP-VAE, it is possible to discover novel seed compounds with effects greater than the original drugs by exploring the vicinity of existing drug compounds. In addition, for the performance comparison of our NP-VAE model to a baseline approach, we conducted the docking experiment using virtual compounds. These compounds were generated by another machine-learning-based molecular generation tool, REINVENT (version 3.0), developed by Blaschke et al.[42]. This model's architecture is based on a recurrent neural network with SMILES representations and is pre-trained on the ChEMBL chemical compounds database[43]. Specifically, these virtual compounds were generated using the reinforcement learning reward of REINVENT with the QEPPI score[44]. For the comparison, we displayed the histogram of the docking scores for the virtual compounds in Fig. 5c. From these docking simulations, the docking score for Gefitinib ranked in the top 15.17% for NP-VAE and in the top 6.78% for REINVENT.

Figure 6 shows the verification of the interaction between the newly generated drug candidate compounds and the target EGFR (PDB code: 4I22[45]). While it is evident that Gefitinib interacts with methionine at position 793, the ligand with the best docking score was found to interact not only with methionine at position 793 but also with arginine at position 841 and asparagine at position 842. In addition, the ligand with the second-best docking score showed interactions with methionine at position 790, cysteine at position 797, and alanine at position 743. In fact,

previous studies on molecular-targeted drugs have reported hydrogen bonding between Osimertinib and EGFR at positions 790 and 793[41], and a covalent bond with Afatinib at position 797 has been reported[46]. The novel compounds obtained in this study were found to demonstrate interactions consistent with these investigations, and it was shown that compound structures other than pyrimidine have a significant impact on the increase in binding strength with the target. The combination of NP-VAE and docking analysis in this method enables the verification of interactions between diverse compound structures and target proteins, and is expected to contribute not only to the discovery of novel pharmaceutical compounds but also to the elucidation of drug mechanisms of action and the acquisition of new insights.

**Computational complexity**. The training of generative models generally demands significant computational time, particularly for models with a large number of parameters like ours, which comprises tens of millions of parameters. Therefore, we conducted a comparison of computational times between NP-VAE and existing methods. Table 4 shows the computational time required for one epoch during training. NP-VAE proved to be faster compared to VAE-based methods such as HierVAE. However, it exhibited longer computational times when compared to non-VAE methods. This discrepancy can be attributed to the fact that all VAE models, including ours, adopt LSTM, a sequential computation process, which appears to be the bottleneck in terms of computational time.

**Synthetic accessibility of natural product-like compounds**. From Table 3, it can be observed that compounds with high NP-likeness scores tend to have even higher synthetic accessibility (SA) scores, which indicates the difficulty of synthesis. Thus, a main issue in the context of handling large and complex compounds resembling natural products is the synthetic accessibility of the generated compounds. Many microbial natural products often possess intricate and unique molecular architectures. Their complexity arises from the diverse and specialized biosynthetic pathways found in nature, and therefore presents a high degree of

synthetic difficulty. Due to their complexity, the synthesis of natural products often requires many steps and translates to challenges in reproducing these compounds in the laboratory. A potential application for our NP-VAE model in addressing the synthetic accessibility challenge is the simplification of compound structures within the chemical latent space developed by NP-VAE. This involves searching for structures that are simpler, smaller in size, of the same bioactivity, and easier to synthesize, in the vicinity of known natural product structures in the chemical latent space. These candidates could then be subjected to further experimental validation. Confirming the effectiveness of this approach is the next crucial challenge.

## Conclusion

We developed a VAE model capable of handling large molecular structures, such as natural compounds, with high accuracy, and constructed a chemical latent space that takes into account both structural and functional information including chirality. By using a large set of pharmaceutical compounds and natural compounds as compound libraries, we successfully constructed the chemical latent space incorporating pharmacological effects, enabling statistical and comprehensive analysis. NP-VAE demonstrated consistent performance as a generative model across various indices. Furthermore, by exploring the chemical latent space, we succeeded in generating novel compound structures with the desired functionality, and demonstrated that *in-silico* selection of drug candidate compounds is possible by combining with docking analysis-based screening.

## Material and methods

**The architecture of the NPVAE**. The overall structure of NP-VAE is shown in Fig. 7. NP-VAE consists of three components: preprocessing, Encoder, and Decoder. In preprocessing, the compound structure is decomposed into fragments according to certain rules and converted into a corresponding tree structure. In the Encoder, the tree structure obtained from preprocessing and the original compound structure are inputted to calculate the latent variable $z$. In the Decoder, taking the latent variable $z$ as input, a tree structure is generated using a depth-first algorithm, and then converted back into the corresponding compound structure. The summary of the specific components in the NP-VAE architecture compared to existing methods such as HierVAE and JT-VAE is as follows.

*Novel components of NP-VAE*
Preprocessing Objectives in NP-VAE: Simplification of compound structure: The first objective is to convert the input compound structure into a simpler structure that can be more easily handled. Particularly when dealing with large molecular structures, calculating at the single-atom level, as JT-VAE does, would result in an enormous order both in time and space. To address this, we devised a procedure to capture compound substructures by decomposing them into several fragments.

Extraction of meaningful physicochemical features: The second objective is to reshape the compounds so that meaningful physicochemical features can be extracted. Aromatic rings like benzene, as well as functional groups deeply involved in the physicochemical properties, such as amide and carboxyl groups, should be treated as a single fragment rather than a sequence of individual atoms. The compound decomposition algorithm was determined based on these objectives.

Chirality handling: We have devised a method of managing and preserving the chirality of molecules, which is an essential factor in the 3D complexity of compounds.

*Components inspired by existing methods*. Variational inference: NP-VAE, like HierVAE and JT-VAE, employs variational inference to learn a continuous latent space.

**The preprocessing of NP-VAE**. There are two objectives in the preprocessing of NP-VAE. The first one is to convert the input compound structure into a simpler structure that can be more easily handled. Particularly when dealing with large molecular structures, calculating at the single-atom level would result in an enormous order both in time and space. To address this, we devised a procedure to capture compound substructures by decomposing them into several fragments. Also, the presence of loop structures in the molecular graph would require a significant computational cost during compound generation in the subsequent Decoder; thus, we aim to capture the structure as a tree without loops. The second objective is to reshape the compounds so that meaningful physicochemical features can be extracted. Aromatic rings like benzene, as well as functional groups deeply involved in the physicochemical properties, such as amide and carboxyl groups, should be treated as a single fragment rather than a sequence of individual atoms. The compound decomposition algorithm was determined based on these objectives.

In the preprocessing step, we first extract substructures fragmented from the entire compound structures according to the decomposition procedure (in Supplementary Methods), and save them as substructure labels while converting them into corresponding tree structures (Supplementary Fig. S5).

When defining the tree structure $\mathcal{T}$ corresponding to the compound structure $G$, the number of nodes in $\mathcal{T}$ matches the number of substructures, and edges are drawn between neighboring substructures within $G$. At each node of $\mathcal{T}$, the ECFP calculated from the corresponding substructure is stored as a feature vector.

**NP-VAE encoder**. In the Encoder, feature extraction of compound structures is performed combining two processes (Supplementary Fig. S6). First, for each ECFP stored in the nodes of the tree structure $\mathcal{T}$, a feature vector $h$ is obtained using Child-Sum Tree-LSTM[22]. Let $C(j)$ be all the child nodes of node $j$, $x_j$ be the ECFP of node $j$, $h_j$ be the hidden state of node $j$ in the Tree-LSTM, $i_j$ be the input gate, $o_j$ be the output gate, $c_j$ be the memory cell, and $f_{jk}$ be the forget gate for child node $k$ of node $j$. The Child-Sum Tree-LSTM is defined by the following equations:

$$h_j = o_j \odot \tanh(c_j) \tag{2}$$

$$o_j = \text{sigmoid}\left(W^o x_j + U^o \widetilde{h}_j + b^o\right) \tag{3}$$

$$\widetilde{h}_j = \sum_{k \in C(j)} h_k \tag{4}$$

$$c_j = i_j \odot u_j + \sum_{k \in C(j)} f_{jk} \odot c_k \tag{5}$$

$$i_j = \text{sigmoid}\left(W^i x_j + U^i \widetilde{h}_j + b^i\right) \tag{6}$$

$$u_j = \tanh\left(W^u x_j + U^u \widetilde{h}_j + b^u\right) \tag{7}$$

$$f_{jk} = \text{sigmoid}\left(W^f x_j + U^f h_k + b^f\right) \tag{8}$$

Here, $\odot$ represents the element-wise product, $W^i$, $W^f$, $W^o$, $W^u$, $U^i$, $U^f$, $U^o$, $U^u$ are the weights learned in the fully connected layers, and $b^i$, $b^f$, $b^o$, $b^u$ are the learned constants (biases).

Second, we compute the ECFP for the entire compound structure. This is denoted as $x_0$, and by inputting it into the $L$-layer fully connected layers, we obtain the output $x_L$. The output $x_L$ is defined by the following formula, with the weights and biases of the $l$-th fully connected layer denoted as $W^l$ and $b^l$, respectively.

$$x_l = W^l x_{l-1} + b^l \, (1 \leq l \leq L) \qquad (9)$$

Lastly, we sum up the feature vector $h_0$, which corresponds to the root node obtained from the Tree-LSTM, and the output $x_L$ of the fully connected layers. Using random noise $\varepsilon \sim N(0, I)$, we compute the latent variable $z$ via the reparameterization trick. With the weights of the fully connected layers denoted as $W^{enc}, W^\mu, W^\sigma$ and biases as $b^{enc}, b^\mu, b^\sigma$, the expression is as follows.

$$z = \mu + \varepsilon \odot \sigma \qquad (10)$$

$$\mu = W^\mu h_G + b^\mu \qquad (11)$$

$$\sigma = W^\sigma h_G + b^\sigma \qquad (12)$$

$$h_G = \left[ W^{enc} \left( h_0 + x_L \right) + b^{enc} \right] \qquad (13)$$

**NP-VAE decoder**. In the Decoder, based on the input latent variable $z$, a tree structure is generated using a depth-first sequential algorithm and is then converted to a compound structure for output (Supplementary Fig. 7). NP-VAE decoder consists of seven procedures: Root label prediction, Topological prediction, Bond prediction, Label prediction, Update the variable $z$, Conversion to compound structure, and Chirality assignment. We briefly describe each procedure, and for a full description of NP-VAE algorithm, see the Supplementary Methods.

In the first step of the Decoder, called Root label prediction, we predict the substructure label that will be assigned to the initially generated root node. The prediction of substructure labels is selected from all the substructure labels obtained during the preprocessing of NP-VAE. The input latent variable $z$ to the Decoder is fed into $L_r$ fully connected layers, and a multi-class classification is performed. In Topological prediction, we predict whether or not to generate a new child node under the current node. If it is predicted to generate a child node, we then proceed to bond prediction and label prediction. On the other hand, if it is predicted not to generate a child node, we terminate the Decoder process if the node is at the root position; otherwise, we backtrack from the current node to its parent node. In Bond prediction, we predict the type of bond between the current node's substructure and the substructure of the newly generated child node. In Label prediction, we predict the substructure label that corresponds to the newly generated child node. After label prediction or backtrack, we compute $z_{t+1}$ from $z_t$ using a fully connected layer. The output $z_{t+1}$ is defined by the following equation, where $W$ and $b$ are the weights and biases of the fully connected layer, respectively.

$$z_{t+1} = \tanh(W(z_t + h_i) + b) \qquad (14)$$

Here, $h_i$ is the feature vector obtained by performing the Child-Sum Tree-LSTM computation, which represents the features at node $i$ after propagating the ECFP stored in each node in the tentative tree structure. During child node generation, the features are transmitted through backward propagation from the root node to the leaf node, and that child node is set as node $i$ (Supplementary Fig. S8a). On the other hand, during backtrack, after the backward propagation from the root node to the leaf node, a forward propagation from the leaf node to the root node

is performed, and the Backtrack destination parent node is set as node $i$ (Supplementary Figure S8(b)). In Conversion to compound structure, after generating the tree structure, the substructure labels of each node are connected and converted into the corresponding compound structure. Since information about the atoms corresponding to the bonding sites within the substructure and their bonding order is already included in the substructure labels, the compound structure can be uniquely determined from the generated tree structure (Supplementary Fig. S8c).

In Assignment of chirality, to handle three-dimensional information of compounds in the Encoder, ECFP with chirality information is used. In the Decoder, the latent variable $z$ is input to the $L_c$-layer fully connected layer, and the predicted ECFP value is output. The output $u_{L_c}$ is defined by the following equation, where the weights and biases of the $l$-th fully connected layer are $W_c^l$ and $b_c^l$, respectively.

$$u_l = \tanh\left( W_c^l u_{l-1} + b_c^l \right) \left( 1 \leq l \leq L_c - 1 \right) \qquad (15)$$

$$u_{L_c} = \text{sigmoid}\left( W_c^{L_c} u_{L_c-1} + b_c^{L_c} \right) \qquad (16)$$

$$u_0 = z \qquad (17)$$

Here, the dimension of $u_{L_c}$ is same as the bit size of ECFP. After the two-dimensional structure of the compound is output based on the aforementioned sequential algorithm, all possible stereoisomers are enumerated and their ECFP is calculated. The Euclidean distance between them and $u_{L_c}$ is computed, and the three-dimensional structure corresponding to the ECFP with the smallest distance is selected as the output compound structure.

**Learning**. During training, to ensure proper learning, even if an incorrect prediction is made in the decoding process that cannot reconstruct the input data, the learning proceeds by propagating feature values on the tree structure, replacing it with the correct one for reconstruction. Additionally, to ensure that the latent space generated by NP-VAE not only accounts for structural information of compounds but also incorporates functional information, such as bioactivity, the latent variable $z$ is input to the $L_p$-layer fully connected layer for predicting the activity value of the input compounds. The output $u_{L_p}$ is defined by the following formula, with the weights and biases of the $l$-th fully connected layer represented by $W_p^l$ and $b_p^l$, respectively.

$$u_l = W_p^l u_{l-1} + b_p^l \left( 1 \leq l \leq L_p \right) \qquad (18)$$

$$u_0 = z \qquad (19)$$

By adding the difference loss between the predicted value $u_{L_p}$ and the true activity value in the loss function, functional information is incorporated into the chemical latent space.

The loss function during NP-VAE training consists of a weighted sum of the cross-entropy loss ($CE$) calculated from each prediction task in the Decoder, the KL divergence ($D_{KL}$) representing the distance between the distribution $Q(z|X)$ of latent variables and the Gaussian distribution, the binary cross-entropy loss ($BCE$) in three-dimensional structure prediction, and the mean squared error ($MSE$) in functional information prediction. Let the ground truth values for Root Label prediction, Topological prediction, Label prediction, and Bond prediction be $y_r$, $y_\tau$, $y_s$, and $y_b$ respectively (represented by a vector where the index of the correct label is 1 and all others are 0), and let the true ECFP value be $y_c$ and the true functional information be $y_p$. The

loss function $L$ is defined as follows:

$$L = \alpha \cdot CE\left(y_r, u_{L_r}\right) + \beta \cdot \sum_i CE\left(y_{\tau,i}, u_\tau\right) + \gamma \cdot \sum_j CE\left(y_{s,j}, u_{L_s}\right)$$
$$+ \delta \cdot \sum_j CE\left(y_{b,j}, u_{L_b}\right) + \varepsilon \cdot BCE\left(y_c, u_{L_c}\right)$$
$$+ \epsilon \cdot MSE\left(y_p, u_{L_p}\right) + \zeta \cdot D_{KL}[Q(z|X)||P(z)] \tag{20}$$

$$CE(y, \hat{y}) = -y \log \hat{y} \tag{21}$$

$$BCE(y, \hat{y}) = -\left[y \log \hat{y} + (1-y) \log(1-\hat{y})\right] \tag{22}$$

$$MSE(y, \hat{y}) = (y - \hat{y})^2 \tag{23}$$

$$D_{KL}[Q(z|X)||P(z)] = -\frac{1}{2}\sum_d \left(1 + \log \sigma_d{}^2 - \mu_d{}^2 - \sigma_d{}^2\right) \tag{24}$$

Here, $\alpha$, $\beta$, $\gamma$, $\delta$, $\varepsilon$, $\epsilon$, and $\zeta$ are hyperparameters used to adjust the contribution of each term.

**Assay for EGFR inhibitory activity of Yessotoxin**. EGFR tyrosine kinase assay was carried out in the same manner as previously reported[40,41]. Briefly, EGFR was obtained as a membrane fraction of A431 cells, and an aliquot of DMSO solution of Yessotoxin or AG1478, a selective EGFR inhibitor and positive control, was added to an HEPES buffer (pH 7.4) containing the A431 membrane fraction, $MnCl_2$, angiotensin II, and EGF. After incubating the mixture at 25 °C for 30 min, the kinase reactions were initiated by the addition of [$\gamma$-$^{32}$P]ATP. The reaction mixture was incubated at 0 °C for 15 min, and then the reaction was stopped by addition of TCA and BSA. After removing precipitated proteins by centrifugation, the radioactivity of the supernatant was counted with a liquid scintillation counter.

**Docking analysis for EGFR**. Protein-ligand docking analysis was conducted using Schrödinger Glide (version 2020-2)[40]. The complex structure of EGFR tyrosine kinase and Gefitinib (PDB ID: 2ITY chain A) was used, from which Gefitinib was removed from the PDB file. The input files for the protein structure were prepared using the Protein Preparation Wizard in Schrödinger Maestro. The ligand was prepared from the SDF file by LigPrep, generating a 3D conformer. All tautomers were generated by LigPrep. As the docking site, a docking grid of 20 Å x 20 Å x 20 Å from the center of Gefitinib in 2ITY chain A was specified. Docking was carried out using Glide SP mode, and the pose with the best Glide SP score was selected for each ligand.

## Data availability
The evaluation dataset and the processed DrugBank dataset used in this study are available at https://github.com/toshikiochiai/NPVAE.

Two representative collections of compound structures within the project dataset, namely collection A and B, are also available at the same site. Most other compound structures in the project dataset are unpublished, and restrictions apply to the availability of these data, which were used under license for the current study and therefore are not publicly available. Data could, however, be available from the authors upon reasonable request. On the other hand, the model parameters of NP-VAE trained on the evaluation dataset and on the drug-and-natural-product dataset are all available at the above site.

The source data for the graphs in Fig. 2(b), Figs. 5(a) and 5(c) are available as Supplementary Data.

## Code availability
The source code for the implementation of NP-VAE is available at https://github.com/toshikiochiai/NPVAE.

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

## Acknowledgements

This work was supported by a Grant-in-Aid for Scientific Research on Innovative Areas "Frontier Research on Chemical Communications" [no. 17H06410, no. 22H04901 (Y.S.)] from the Ministry of Education, Culture, Sports, Science and Technology of Japan. This work was also supported by Grant-in-Aid for Transformative Research Area (A) "Latent Chemical Space" [23H04887 (M.O.), 23H04881 (K.K., N.M.), 23H04885 (Y.S.), and 23H04880 (M.O., K.K., Y.S.)] from the Ministry of Education, Culture, Sports, Science and Technology, Japan.

We are grateful to the project members in "Frontier Research on Chemical Communications" for contributing to construction of the database for the original compound library.

## Author contributions

T.O.; implemented the software, analysed data, and compared with the existing methods. T.I.; compared with the existing methods and analysed data. M.A.; analysed data. K.F., M.O.; performed docking analysis. N.M.; performed EGFR inhibitory assay. S.I., M.U., T.S.; set up the database for the original compound library. H.K., K.K.; set up the database for the original compound library and edited the paper. Y.S.; designed and supervised the research, analysed data, and wrote the paper. All authors read and approved the final manuscript.

## Competing interests

The authors declare no competing interests.
