## [Peer Review File · Communications Chemistry]

Reviewers' comments:

Reviewer #1 (Remarks to the Author):

The authors present a novel generative model for the generation of large molecules resembling natural products based on variational autoencoders (VAE). In the methods and the supplementary material the authors present a detailed description of the architecture of their VAE. As basic representation of molecules a substructure-based tree representation is utilized that is obtained by a systematic fragmentation process during encoding, and correspondingly generated using a depth-first strategy during decoding. The method description (including the supplementary material) are comprehensive and well written. A noteworthy addition to the generative model is the inclusion of chirality, which is achieved by considering chirality-sensitive fingerprint representations.

* A detail that is lacking from the method description is how the NP-score was integrated in the representations for the learning process.

* As common with generative models, performance evaluation is very difficult. Some of the performance metrics reported in the manuscript are not meaningful. In Table 1, validity is reported. The tree-based approach used here will, by design, only yield valid molecular structures (similar to the other tree-based methods). The measure is thus only relevant for generative approaches using linear representations like Smiles. Furthermore, validity is not a crucial metric per se as invalid representations as a generative model can be easily augmented to filter out invalid representation.

* Table 2 reports Frechet ChemNet distance (FCD), synthetic accessibility (SA), and QED score. Here, I would consider the FCD most meaningless given that natural products would fall outside the applicability domain of the ChemNet model. Thus a high FCD should be expected. However a high FCD can also be an indication of bad generative performance like mode collapse. Furthermore, the SA score reported seems very low, given that natural product like molecules would be expected to have bad synthetic accessibility. Likewise QED scores for these larger molecules might be expected to be lower and not necessarily an indication of good or bad performance.

Based on these considerations, I find the conclusion on lines 242f: "These findings confirm that NP-VAE can generate novel compounds with chemical and biological properties similar to real molecules compared to SM-RNN and with characteristics closer to the training data compared to MoFlow." is unfounded.

In addition to uniqueness and novelty, the authors should consider determining distributions of physico-chemical properties of generated models and compare them to those of the target chemical space (e.g.. natural products.) The paper <https://doi.org/10.1038/s42256-021-00368-1> by Skinnider et al. lists a variety of properties that should be considered.

* While generative models for larger natural product-like molecules is an interesting concept, the paper does not go beyond the hypothetical identification of novel molecules in docking studies. Synthetic accessibility of these proposed molecules might prove a crucial obstacle in any practical application. I would expect the manuscript to critically address this issue in the discussion and outline potential use cases for their model.

In summary, the manuscript might be considered for publication after major revisions.

Reviewer #2 (Remarks to the Author):

Summary

This work sets out to build a molecular autoencoder that can build a latent space for large molecules and natural products which

To do this, the authors propose a new graph-based Variational Autoencoder which decomposes the input structure into substructures and builds a tree to encode the connectivity and build a latent feature vector.

The model is trained to reconstruct the input structure from the latent feature vector.

The VAE is jointly trained on some property prediction tasks to shape the embedding space with functional information and not just chemical and structural information.

Experiments show that the model outperforms state of the art baselines on standard datasets in various generative model metrics.

Next, the final model is trained on DrugBank and natural products databases and some visualizations, interpolation experiments and molecular docking demonstrate the usefulness of the learned space.

The aim of pushing VAEs towards more realistic and biologically significant regions of chemical space is worthwhile.

I appreciate the types of case studies that the authors present (docking, binding assays, interpolation, visualization) however I was not convinced at a quantitative level that this model clearly outperforms existing architectures.

For this reason I am unable to recommend the work in its current form for publication.

See below for details.

Major Remarks

- * Needs discussion of runtime efficiency and computational complexity for encoding and decoding.
- * Why were the baseline models (JTVAE, HierVAE, etc.) used in Table 1 not also used in Table 2?
- * In both Table 1 and 2 the variance of the means should be included (especially since the decoding was repeated 100 times, I presume that one could easily report the variability in the measurements)
- * The natural products and DrugBank training in Table 3 only contains results from NP-VAE (proposed model). The authors should also train existing architectures to show that their architecture is what contributes to the performance and not just a change of training data.
- * The question of generating large molecules which was a starting motivation was never directly tackled. A plot showing size distributions of the molecules generated by this method vs others would be good to

see.

- * The subsequent results are mainly qualitative when they could have been made quantitative. For example:
- * The smoothness of the NP-likeness score with respect to embedding distance can be plotted as a correlation with a quantitative regression score. Instead it is just left for visual inspection on a t-SNE which is often not reproducible. The same can be done for the interpolation experiment.
- * The Bayesian Optimization experiment can be repeated for many points in the space and an average improvement in the objective function reported.
- * Most importantly, none of the experiments mentioned in the previous point were compared to other methods when they could have been (as far as I can tell).
- * The docking experiment is interesting but has no baseline. How does the sampling in this latent space compare to random sampling? Or sampling in the latent space of another pre-trained model?
- * In short: because none of the applications are uniquely possible with this method (any VAE could be used to do this) one needs to see that this VAE can do it better than others by direct and quantitative comparison, or to clearly discuss why it is not possible with existing methods.

Minor remarks

- * Provide some intuition for how the NP scores work in the main text.
- * Could not install dependency for mkl-service with pip or conda. Please include an automated installation system (e.g. requirements.txt or environment.yml)
- * In table 3 it's not clear what the Tanimoto coefficient represents. This is a distance on bit vectors. Do the authors compute a fingerprint for the input and output compounds to do this? Also would this not be 1 if the model memorizes the inputs?
- * Line 160 "Despite this, the latest state-of-the-arts methods have not conducted any evaluation of the generalization ability at all [3,9-11]." I'm not sure I understand what is meant by this. Existing models test their reconstruction on held-out data. In fact, the values for the baseline in Table 1 on generalization ability comes directly from the HierVAE paper. Please clarify.
- * When describing the architecture of the VAE it is important to have existing methods in mind for the various algorithmic choices. For example, HierVAE and JT-VAE also use functional substructures in the encoding so it would be good for the reader to know which specific components of the algorithm lead to solving the initial problem and which are borrowed from existing work.
- * There are quite a few papers on functional optimization with latent space sampling which should be discussed (e.g. Boitreau, Jacques, et al. "OptiMol: optimization of binding affinities in chemical space for drug discovery." *Journal of Chemical Information and Modeling* 60.12 (2020): 5658-5666., Griffiths, Ryan-Rhys, and José Miguel Hernández-Lobato. "Constrained Bayesian optimization for automatic chemical design using variational autoencoders." *Chemical science* 11.2 (2020): 577-586., Korovina, Ksenia, et al. "Chembo: Bayesian optimization of small organic molecules with synthesizable recommendations." *International Conference on Artificial Intelligence and Statistics*. PMLR, 2020.)
- * Fig 1(a) and (b) label the colorbar
- * Fig 5. The x-ticks are very hard to read. Plot instead just the midpoint of the bin and maybe omit every other one.

Dear Reviewers,

We greatly appreciate the reviewers' constructive comments, which helped us to considerably improve the quality of our paper. The manuscript has been revised accordingly. The revisions in the main text are highlighted in red. Here, we present our point-by-point responses to each of the reviewers' comments along with details of all the revisions made.

In summary, the main revisions include the following:

- We conducted additional performance evaluations of existing models, including baseline models such as HierVAE, to ensure a consistent and comprehensive comparison across all evaluated metrics, including a variety of indices proposed in the benchmarking studies. The results of these evaluations are summarized in the new Table 3 in the revised manuscript.
- We acknowledge that our primary motivation was to develop a VAE model capable of handling large and complex molecules for the construction of chemical latent space, not necessarily focused on generating large molecules. In this sense, our previous title is somewhat misleading. Therefore, we changed the title to "VAE-based model of constructing chemical latent space for large molecular structures with 3D complexity". Along with this change, we have appropriately revised relevant statements in the Abstract and Introduction.

[Comment of Reviewer #1:]

A detail that is lacking from the method description is how the NP-score was integrated in the representations for the learning process.

Response: Thank you very much for pointing out this important issue. As we mentioned in the subsection "Construction of chemical latent space with natural compounds", the NP-likeness score was incorporated in the training process as functional information. The functional information is incorporated into the learning process as a component of the loss function, that is, the mean squared error in functional information prediction. The details are described in the subsection "Learning" and the formulas (20) and (23) of the manuscript. To make it clearer, we have added the following statement to the subsection "Construction of chemical latent space with natural compounds":

"More precisely, the NP-likeness score is incorporated into the learning process as functional information, which is implemented by a component of the loss function."

[Comment of Reviewer #1:]

As common with generative models, performance evaluation is very difficult. Some of the performance metrics reported in the manuscript are not meaningful. In Table 1, validity is reported. The tree-based approach used here will, by design, only yield valid molecular structures (similar to the other tree-based methods). The measure is thus only relevant for generative approaches using linear representations like Smiles. Furthermore, validity is not a crucial metric per se as invalid representations as a generative model can be easily augmented to filter out invalid representation.

Response:

Thank you very much for pointing out this issue. Your point is well taken. Regarding the presentation in Table 1, we followed the format used in previous studies. In conjunction with your next comment, we have conducted the performance evaluation of the generative model by incorporating additional metrics, which were summarized in the new Table 3 in the revised manuscript.

[Comment of Reviewer #1:]

Table 2 reports Frechet ChemNet distance (FCD), synthetic accessibility (SA), and QED score. Here, I would consider the FCD most meaningless given that natural products would fall outside the applicability domain of the ChemNet model. Thus a high FCD should be expected. However a high FCD can also be an indication of bad generative performance like mode collapse. Furthermore, the SA score reported seems very low, given that natural product like molecules would be expected to have bad synthetic accessibility. Likewise QED scores for these larger molecules might be expected to be lower and not necessarily an indication of good or bad performance.

Based on these considerations, I find the conclusion on lines 242f: “These findings confirm that NP-VAE can generate novel compounds with chemical and biological properties similar to real molecules compared to SM-RNN and with characteristics closer to the training data compared to MoFlow.” is unfounded.

In addition to uniqueness and novelty, the authors should consider determining distributions of physico-chemical properties of generated models and compare them to those of the target chemical space (e.g. natural products.) The paper <https://doi.org/10.1038/s42256-021-00368-1> by Skinnider et al. lists a variety of properties that should be considered.

Response:

Thank you very much for your valuable comments. We recognized the importance of selecting meaningful metrics that are relevant to the domain of natural products. Here is our response to your comments:

Firstly, we would like to mention that all the models being compared here were trained on the same dataset. As stated in the main text, this dataset comprises compounds from DrugBank, including small molecules, as well as compounds primarily consisting of natural products from the project. Given that machine learning models heavily depend on the input training data, it’s not the case that our model uniquely generates a large number of natural product-like molecules.

Secondly, we included FCD, SA and QED scores as they are commonly used metrics in the literature, but we understand the limitations they might have in our specific case. In light of your feedback, we incorporated a variety of metrics proposed in benchmarking studies such as MOSES [29] and GuacaMol [30]. These metrics were also discussed in the paper by Skinnider et al., which you kindly pointed out. We have evaluated our model based on its generative capabilities in terms of these various properties. We have excluded the FCD score from the evaluation. The additional result is described in the new Table 3 in the revised manuscript.

We understand your concern regarding the conclusion statement on lines 242f. Based on your feedback, we removed this statement from the main text to avoid any confusion.

[Comment of Reviewer #1:]

While generative models for larger natural product-like molecules is an interesting concept, the paper does not go beyond the hypothetical identification of novel molecules in docking studies. Synthetic accessibility of these proposed molecules might prove a crucial obstacle in any practical application. I would expect the manuscript to critically address this issue in the discussion and outline potential use cases for their model.

Response:

We recognize the importance of synthetic accessibility in the context of generative models for chemical structures, especially for large and complex molecules resembling natural products. In the revised manuscript, we included a thorough discussion in the Discussion where we critically addressed the synthetic accessibility of the molecules generated by our model. We appreciate your suggestion to outline potential use cases for our model. In the revised manuscript, we included a discussion of potential use cases where our model could be beneficial. For example, we discussed how our model could be used in the simplification of compound structures within the chemical latent space developed by NP-VAE. This involves searching for structures that are simpler, smaller in size, and easier to synthesize in the space.

[Comment of Reviewer #2:]

Major Remarks

Needs discussion of runtime efficiency and computational complexity for encoding and decoding.

Response:

Thank you very much for pointing out this important issue. We added a new subsection “Computational complexity” on the runtime efficiency of our model NP-VAE and also conducted a comparison of computation time with other methods.

[Comment of Reviewer #2:]

Why were the baseline models (JTVAE, HierVAE, etc.) used in Table 1 not also used in Table 2?

Response:

Thank you for your insightful question. The baseline models (JTVAE, HierVAE, etc.) were initially used in Table 1 (in the previous version of the manuscript) to provide a direct comparison of our method with established models in terms of reconstruction accuracy. In Table 2 (in the previous version of the manuscript), we aimed to focus on more specific and advanced metrics that are particularly relevant to the aspects of the generative model. We understand that including the baseline models in Table 2 as well would provide a more comprehensive comparison. In light of your comment, we revised Table 2 (the new Table 3 in the revised manuscript) to include the results for the baseline models for a more consistent and thorough comparison across all evaluated metrics. Please note that regarding JTVAE, due to its somewhat outdated implementation, it took an impractical amount of computation time and did not complete the calculations even with the restricted dataset. In addition, HierVAE can be considered an improved version of JTVAE, developed by the same authors, so comparing the accuracy of HierVAE should be sufficient.

[Comment of Reviewer #2:]

In both Table 1 and 2 the variance of the means should be included (especially since the decoding was repeated 100 times, I presume that one could easily report the variability in the measurements)

Response:

Thank you for pointing this out. Regarding the presentation in Table 1, we followed the format used in previous studies. Furthermore, upon inspecting the code of HierVAE, it was found that no noise was added during the generation iterations. As a result, it is not possible to calculate the variance for HierVAE. Additionally, in many other VAE model studies, it is observed that noise is not added in the accuracy evaluation. Since accuracy tends to be higher without the addition of noise, NPVAE is at a disadvantage as it calculates reconstruction accuracy with added noise, but it still surpasses in accuracy. Therefore, we decided to present Table 1 in its current format.

[Comment of Reviewer #2:]

The natural products and DrugBank training in Table 3 only contains results from NP-VAE (proposed model). The authors should also train existing architectures to show that their architecture is what contributes to the performance and not just a change of training data.

Response:

Thank you for this valuable suggestion. We understand the importance of demonstrating that the performance improvements are attributable to our proposed architecture, and not solely due to the change in training data. On the other hand, the state-of-the-art VAE models including JT-VAE and HierVAE were unable to handle data for compounds of this larger size in the drug-and-natural-product dataset. This fact was already mentioned in the first version of our manuscript. In the revised manuscript, we have made this point more explicit. Therefore, we had to prepare a restricted dataset where all existing methods can be executed. This restricted dataset was constructed by first reducing the drug-and-natural-product dataset to compounds with fewer than 100 non-hydrogen atoms and further removing some compounds that caused errors with HierVAE. The results of comparing the existing models on the restricted dataset are presented in the new Table 3 in the revised manuscript.

[Comment of Reviewer #2:]

The question of generating large molecules which was a starting motivation was never directly tackled. A plot showing size distributions of the molecules generated by this method vs others would be good to see.

Response:

First, we would like to acknowledge that our initial motivation was to develop a VAE model capable of handling large molecules, not necessarily focused on generating large molecules. In addition, machine learning heavily depends on the input training data, so among models trained on the same dataset, it does not necessarily mean that only our model can generate a large number of compounds with larger sizes. Therefore, we added the new Table 2 in the revised manuscript, to show the differences in the maximum number of

atoms and molecular weight of the compounds that our model can handle compared to existing VAE models. Second, to respond to your suggestion, we created a plot that illustrated the size distributions of the molecules generated by our method compared to those generated by other methods in the Supplementary Figure S2 in the revised manuscript.

[Comment of Reviewer #2:]

The subsequent results are mainly qualitative when they could have been made quantitative. For example:

* The smoothness of the NP-likeness score with respect to embedding distance can be plotted as a correlation with a quantitative regression score. Instead it is just left for visual inspection on a t-SNE which is often not reproducible. The same can be done for the interpolation experiment.

* The Bayesian Optimization experiment can be repeated for many points in the space and an average improvement in the objective function reported.

Response:

Thank you for your constructive feedback. We agree that providing quantitative results, in addition to qualitative observations, would strengthen the rigor and reproducibility of our findings. In response to your comment:

We computed and reported a quantitative correlation score (the Pearson correlation coefficient) between these two variables. We included this analysis with the new Supplementary Figure S3 for the plot in the revised manuscript. This analysis should also provide insights into the interpolation experiment. In addition, we have ceased using the term “continuity” to avoid any misunderstandings.

Regarding the Bayesian Optimization experiment, we repeated the experiment for multiple points in the latent space and reported the average improvement in the objective function, along with the standard deviation to capture the variability. This additional analysis was included in the revised manuscript.

[Comment of Reviewer #2:]

Most importantly, none of the experiments mentioned in the previous point were compared to other methods when they could have been (as far as I can tell).

Response:

Thank you for emphasizing this critical point. We understand the importance of comparative evaluation with other methods to clearly demonstrate the advantages and potential limitations of our proposed model. In response to your comment, and as indicated in our responses to your previous comments:

We extended our analysis to include comparisons with established methods, such as HierVAE, across all the experiments mentioned. For each experiment, we reported the same quantitative metrics for our method and the comparison methods, as shown in the new Table 3 in the revised manuscript. Once again, we would like to emphasize that our primary motivation was to develop a VAE model capable of handling large and complex molecules for the construction of chemical latent space. We added the new Table 2 to show the differences in the sizes of compounds that our model can handle compared to other VAE models.

[Comment of Reviewer #2:]

The docking experiment is interesting but has no baseline. How does the sampling in this latent space compare to random sampling? Or sampling in the latent space of another pre-trained model?

Response:

Thank you for pointing out the need for a baseline in the docking experiment. In response to your comment: We introduced a baseline comparison in the docking experiment. Specifically, we compared the performance of our model to a baseline approach, which involved conducting the docking experiment using virtual compounds generated by another machine-learning-based molecular generation tool, REINVENT (version 3.0), developed by Blaschke et al. [43] and well used in the community. This model's architecture is based on a recurrent neural network with SMILES representations and is pre-trained on the ChEMBL chemical compounds database [44]. For the comparison, we reported key metrics such as the binding affinity scores and the percentage of molecules that meet certain binding criteria as shown in Figure 5 (a) and (c). From these docking simulations, the docking score for Gefitinib ranked in the top 15.17% for NP-VAE and in the top 6.78% for REINVENT. This provided a quantitative measure to assess the effectiveness of sampling in our model's latent space compared to the baseline approach.

[Comment of Reviewer #2:]

In short: because none of the applications are uniquely possible with this method (any VAE could be used to do this) one needs to see that this VAE can do it better than others by direct and quantitative comparison, or to clearly discuss why it is not possible with existing methods.

Response:

We conducted extensive direct and quantitative comparisons between our NP-VAE model and existing VAE models, HierVAE, across various metrics, as responded to several previous comments of Reviewer#2. Once again note that regarding JTVAE, due to its somewhat outdated implementation, it took an impractical amount of computation time and did not complete the calculations even with the restricted dataset.

As a result, we were able to demonstrate some advantages of our NPVAE over existing VAE models in the following four aspects:

- It can handle larger and complex molecules.
- It can reconstruction accuracy necessary for construction of accurate chemical latent space.
- It is capable of handling 3D molecular structures, including chirality.
- It demonstrated stable performance as a generative model across various indices.
- It has faster computation times than other VAE models.

The results of these comparisons were presented in the revised manuscript through additional tables and/or figures.

[Comment of Reviewer #2:]

Minor remarks

Provide some intuition for how the NP scores work in the main text.

Response:

Thank you for your suggestion. In response to your comment, we revised the main text to include a more detailed and accessible explanation of the NP-likeness scores. Specifically:

We explained that the NP-likeness score is a measure designed to estimate how closely a given molecule resembles known natural products. It is based on the distribution of fragments (substructures) in the molecule compared to a reference set of known natural products.

We briefly described how we used these scores in our model: The NP-likeness scores are incorporated into the learning process as functional information, which is implemented by a component of the loss function. This encourages our model to generate molecules that are more ‘natural product-like.’

[Comment of Reviewer #2:]

Could not install dependency for mkl-service with pip or conda. Please include an automated installation system (e.g. requirements.txt or environment.yml)

Response:

Thank you for bringing this to our attention. We apologize for any inconvenience that the installation process may have caused. We understand the importance of making the installation process as straightforward as possible for users.

In response to your comment, we have created an automated installation system with an environment.yml file on our github site.

We will continue to monitor user feedback regarding the installation process and make further improvements as necessary. We also welcome direct inquiries from users who encounter issues, and we are dedicated to providing timely and effective support.

[Comment of Reviewer #2:]

In table 3 it's not clear what the Tanimoto coefficient represents. This is a distance on bit vectors. Do the authors compute a fingerprint for the input and output compounds to do this? Also would this not be 1 if the model memorizes the inputs?

Response:

Thank you for your insightful questions regarding the Tanimoto coefficient in Table 3. We apologize for any confusion that this might have caused. To clarify:

The Tanimoto coefficient in Table 3 (in the previous version of the manuscript) was used to measure the similarity between the molecular fingerprints of the generated compounds and those of the target compounds in our dataset. Regarding your question about the model memorizing the inputs, we designed our experiments to ensure that the model generalizes well and does not simply memorize the training data. The Tanimoto coefficient was used to assess the similarity between generated compounds and target compounds, not the exact replication of input data. In any case, we removed the statement and result using the Tanimoto coefficient in the main text to avoid any confusion.

[Comment of Reviewer #2:]

Line 160 “Despite this, the latest state-of-the-arts methods have not conducted any evaluation of the generalization ability at all [3,9-11].” I’m not sure I understand what is meant by this. Existing models test their reconstruction on held-out data. In fact, the values for the baseline in Table 1 on generalization ability comes directly from the HierVAE paper. Please clarify.

Response:

Thank you for your question and for giving us the opportunity to clarify this point. We apologize for any confusion that this statement may have caused. In this context, when we refer to the ‘latest state-of-the-arts methods,’ we are specifically referring to the methods cited in references [3,9-11], as indicated. Furthermore, our statement is meant to highlight that the methods cited in references [3,9-11] have not conducted evaluations of their generalization ability using held-out data. To avoid any confusion, we removed this statement in the main text.

[Comment of Reviewer #2:]

When describing the architecture of the VAE it is important to have existing methods in mind for the various algorithmic choices. For example, HierVAE and JT-VAE also use functional substructures in the encoding so it would be good for the reader to know which specific components of the algorithm lead to solving the initial problem and which are borrowed from existing work.

Response:

Thank you for your insightful comment. Regarding the specific components of our NP-VAE, we had detailed information in the Supplementary methods section. In response to your comment, we have added a summary of these components in the main text to provide a comparison between our proposed VAE architecture and existing methods such as HierVAE and JT-VAE. Here is a more detailed breakdown:

Novel Components of Our Algorithm:

- Preprocessing Objectives in NP-VAE:

Simplification of Compound Structure: The first objective is to convert the input compound structure into a simpler structure that can be more easily handled. Particularly when dealing with large molecular structures, calculating at the single-atom level, as JT-VAE does, would result in an enormous order both in time and space. To address this, we devised a procedure to capture compound substructures by decomposing them into several fragments.

- Extraction of Meaningful Physicochemical Features: The second objective is to reshape the compounds so that meaningful physicochemical features can be extracted. Aromatic rings like benzene, as well as functional groups deeply involved in the physicochemical properties, such as amide and carboxyl groups, should be treated as a single fragment rather than a sequence of individual atoms. The compound decomposition algorithm was determined based on these objectives.

- Chirality handling: We have devised a method of managing and preserving the chirality of compounds, which is a crucial factor in the 3D complexity of compounds.

Components Inspired by Existing Methods:

Variational Inference: Our method, like HierVAE and JT-VAE, employs variational inference to learn a continuous latent space.

[Comment of Reviewer #2:]

There are quite a few papers on functional optimization with latent space sampling which should be discussed (e.g. Boitreaud, Jacques, et al. “OptiMol: optimization of binding affinities in chemical space for drug discovery.” *Journal of Chemical Information and Modeling* 60.12 (2020): 5658-5666., Griffiths, Ryan-Rhys, and José Miguel Hernández-Lobato. “Constrained Bayesian optimization for automatic chemical design using variational autoencoders.” *Chemical science* 11.2 (2020): 577-586., Korovina, Ksenia, et al. “Chembo: Bayesian optimization of small organic molecules with synthesizable recommendations.” *International Conference on Artificial Intelligence and Statistics*. PMLR, 2020.)

Response:

Thank you for pointing out these relevant works. In response to your comment, we revised our manuscript to include a discussion of these papers and clarify the distinctions between our work and these existing methods.

For Boitreaud, Jacques, et al. “OptiMol: optimization of binding affinities in chemical space for drug discovery”, we highlighted that OptiMol focuses on an optimization strategy and targets a specific aspect of drug discovery (binding affinities), whereas our NP-VAE model aims to deal with large, complex molecules with 3D structures and desired properties.

For Griffiths, Ryan-Rhys, and José Miguel Hernández-Lobato. “Constrained Bayesian optimization for automatic chemical design using variational autoencoders”, we emphasized that their work primarily uses SMILES representations, which can lead to invalid outputs, making the handling of such outputs a significant focus of their work. In contrast, our NP-VAE model is designed to handle large and complex 3D molecular structures, which is a different problem setting.

For Korovina, Ksenia, et al. “Chembo: Bayesian optimization of small organic molecules with synthesizable recommendations”, this introduces a novel method for synthesizable recommendations beyond the SA score, which represents a distinct challenge compared to our NP-VAE model for handling large molecules with complex 3D structures. We acknowledged the potential value of the synthesizable recommendation method proposed in this paper and discussed our intention to consider incorporating similar strategies in future iterations of our model.

[Comment of Reviewer #2:]

Fig 5. The x-ticks are very hard to read. Plot instead just the midpoint of the bin and maybe omit every other one.

Response:

Thank you for your feedback on the readability of the x-ticks in Figure 5. We adjusted the x-ticks to display only the midpoints of the bins. This simplified the x-axis and made it easier for readers to understand the range that each tick represents.

REVIEWERS' COMMENTS:

Reviewer #1 (Remarks to the Author):

The authors have adequately addressed all issues raised by the reviewers making the publication acceptable in its current form. The font sizes used in some figures, particularly figs. 1a-c are very small and should be increased prior to publication.

Reviewer #2 (Remarks to the Author):

Thank you to the authors for addressing my comments which have been addressed successfully.

My only small remark is that grammatically the title is somewhat awkward. I would simplify to something like this:

"VAE-based latent chemical space for large molecular structures with 3D complexity."

Dear Reviewers,

We appreciate the reviewer's further comments. The manuscript has been revised accordingly. The revisions in the main text are highlighted in red.

[Comment of Reviewer #1:]

The authors have adequately addressed all issues raised by the reviewers making the publication acceptable in its current form. The font sizes used in some figures, particularly figs. 1a-c are very small and should be increased prior to publication.

Response: Thank you very much for pointing out this issue. The font in the figures, especially in Figure 1, has been increased for improved readability.

[Comment of Reviewer #2:]

Thank you to the authors for addressing my comments which have been addressed successfully.

My only small remark is that grammatically the title is somewhat awkward. I would simplify to something like this:

"VAE-based latent chemical space for large molecular structures with 3D complexity."

Response:

Thank you very much for the suggestion regarding the title modification. The manuscript title has been changed to "VAE-based chemical latent space for large molecular structures with 3D complexity", while the reviewer suggested the term "latent chemical space". The reason for this is that the term "chemical latent space" is used in many places in the main manuscript, and previous research papers use the term "molecular latent space".

We hope our manuscript is now suitable for publication.